# Closing the Gap between Inpatient and Outpatient Settings: Integrating Pulmonary Rehabilitation and Technological Advances in the Comprehensive Management of Frail Patients

**DOI:** 10.3390/ijerph19159150

**Published:** 2022-07-27

**Authors:** Lorenzo Lippi, Francesco D’Abrosca, Arianna Folli, Alberto Dal Molin, Stefano Moalli, Antonio Maconi, Antonio Ammendolia, Alessandro de Sire, Marco Invernizzi

**Affiliations:** 1Physical and Rehabilitative Medicine, Department of Health Sciences, University of Eastern Piedmont “A. Avogadro”, 28100 Novara, Italy; lorenzolippi.mt@gmail.com (L.L.); fradabro@gmail.com (F.D.); arianna.folli23@gmail.com (A.F.); stefano.moalli@libero.it (S.M.); 2Dipartimento Attività Integrate Ricerca e Innovazione (DAIRI), Translational Medicine, Azienda Ospedaliera SS. Antonio e Biagio e Cesare Arrigo, 15121 Alessandria, Italy; amaconi@ospedale.al.it; 3Department of Translational Medicine, University of Eastern Piedmont “A. Avogadro”, 28100 Novara, Italy; alberto.dalmolin@med.uniupo.it; 4Health Professions’ Direction, “Ospedale Maggiore della Carità” University Hospital, 28100 Novara, Italy; 5Physical and Rehabilitative Medicine Unit, Department of Medical and Surgical Sciences, University of Catanzaro “Magna Graecia”, Viale Europa, 88100 Catanzaro, Italy; ammendolia@unicz.it (A.A.); alessandro.desire@unicz.it (A.d.S.)

**Keywords:** pulmonary rehabilitation, frailty, telemedicine, outpatient, inpatient, physiotherapy

## Abstract

Pulmonary rehabilitation (PR) is a well-established intervention supported by strong evidence that is used to treat patients affected by chronic respiratory diseases. However, several barriers still affect its spreading in rehabilitation clinical practices. Although chronic respiratory diseases are common age-related disorders, there is still a gap of knowledge regarding the implementation of sustainable strategies integrating PR in the rehabilitation management of frail patients at high risk of respiratory complications. Therefore, in the present study, we characterized the effects of PR in frail patients, highlighting the evidence supporting its role in improving the complex rehabilitative management of these patients. Moreover, we propose a novel organizational model promoting PR programs for frail patients in both inpatient and outpatient settings. Our model emphasizes the role of interdisciplinary care, specifically tailored to patients and environmental characteristics. In this scenario, cutting-edge technology and telemedicine solutions might be implemented as safe and sustainable strategies filling the gap between inpatient and outpatient settings. Future research should focus on large-scale sustainable interventions to improve the quality of life and global health of frail patients. Moreover, evidence-based therapeutic paths should be promoted and taught in training courses promoting multiprofessional PR knowledge to increase awareness and better address its delivery in frail patients.

## 1. Introduction

Pulmonary rehabilitation (PR) is currently considered a milestone in the therapeutic management of patients with chronic pulmonary diseases [1,2]. To date, level one evidence supports its benefits in chronic obstructive pulmonary diseases (COPD), since it improves symptoms, exercise tolerance, physical and psychosocial issues, health-related quality of life, and reduces hospitalizations and socio-sanitary costs [1,2,3,4,5,6].

Although the effects of PR are widely documented, its integration into the comprehensive management of COPD and its accessibility for the patients remains still challenging [7,8]. On the other hand, in recent years, growing evidence emphasized the positive effects of PR in frail patients suffering from disabling conditions including chronic heart disease, metabolic syndrome, musculoskeletal or neurological co-morbidities, and cancer [9,10]. Although chronic diseases and frailty are often treated as separated conditions, they share different multilevel interactions, and several disabling conditions play a pivotal role in frailty onset [11]. Due to the higher rate of pulmonary complications, frail patients might require an integrated rehabilitation approach including PR with relevant implications in terms of both physical wellbeing and quality of life improvement [12,13,14,15,16]. However, to the best of our knowledge, no previous study emphasized the role of PR in these patients. Moreover, the optimal PR protocols are far from being fully characterized, with recent research now focusing on cutting-edge technologies overcoming barriers to PR delivery and improving the already documented cost-effectiveness of a tailored intervention. Lastly, sustainable strategies and organizational models filling the gap between inpatient and outpatient rehabilitation settings were previously never proposed for frail patients, with a growing number of reports that mainly focus on COPD patients only.

In light of this gap of knowledge in the current literature, this narrative review aimed at characterizing the need for PR in frail patients, highlighting the evidence supporting this comprehensive approach to improve not only functional and overall health status, but also reducing sanitary costs due to pulmonary diseases and rehospitalization. Moreover, we sought to propose sustainable strategies to overcome the barriers to PR delivery and increase awareness of both caregiver and health care professions, increasing a broaden transversal knowledge of the PR needs of frail elderly patients.

## 2. Pulmonary Rehabilitation: State of the Art

Pulmonary rehabilitation was defined in 2013 as a “comprehensive intervention based on a thorough patient assessment followed by patient-tailored therapies, which include, but are not limited to, exercise training, education and behaviour change, designed to improve the physical and psychological condition of patients with chronic respiratory disease and to promote the long-term adherence of health-enhancing behaviours” [1].

In a Cochrane Review in 2015, McCarthy and colleagues [6] stated that “additional Randomized Controlled Trials (RCT)s comparing PR and conventional care in COPD” would no longer be justified, so PR should definitely be considered a “standard of care” for patients affected by a chronic respiratory disease, similar to any other well-established pharmacological therapy, oxygen supplementation, and noninvasive ventilation. Moreover, the GOLD (Global Initiative for Chronic Obstructive Lung Disease) annual report stated that “non-pharmacological treatment is complementary to pharmacological treatment and should form part of the comprehensive management of COPD”. Evidence suggests that more symptomatic patients with a higher risk of hospitalizations, as well as patients with worse health status and physical performance (GOLD stage B, C, and D), should be prioritized in a structured program since they might experience the greatest benefits from PR [17]. Moreover, in recent years, growing evidence supported PR in improving outcomes in several respiratory conditions including bronchiectasis [18,19], interstitial lung diseases [20], pulmonary hypertension [21], and recently post-covid physical and respiratory disorders [22,23].

In 2019, an international conference of experts achieved a consensus on the essential components and outcomes of PR, highlighting well-established practices supported by strong evidence in the framework of assessment, contents of the PR program, methods of delivery, and quality assurance. Interestingly, the authors emphasized the role of an endurance and resistance exercise program individually prescribed and progressed, while a pool of “desiderable” components was recognized as useful and potentially integrable into a comprehensive rehabilitation approach. More in detail, upper limbs training, individualized and structured education interventions, action plans for frequent exacerbators, and airway clearance techniques were proposed in the pool of “desiderable” components [24].

In the last years, several trials also showed the safety and the validity of alternative exercise training strategies (such as interval training, water-based exercise training, TaiChi, single-leg exercises) for different phenotypes of patients suffering from chronic respiratory diseases other than COPD, with intriguing implications in patients having co-morbidities that markedly affect functional capacity (i.e., chronic heart disease, metabolic syndrome, musculoskeletal or neurological co-morbidities, and many types of cancer) [25].

## 3. Pulmonary Rehabilitation in Frail Patients with Functional Impairment

In the past few years, the increasing aging of the population coupled with a higher prevalence of age-related diseases led to a growing focus on the disabling complications affecting the pulmonary system [26,27]. Interestingly, it is reported that structural changes in the thoracic cage related to rib cage calcification and age-related kyphosis might decrease pulmonary expansion in the elderly, while a decrease in respiratory muscle function and cough strength might decrease pulmonary dynamics and frailty [28]. Moreover, the chronic inflammation characterizing frailty states and age-related attenuation of immunity response might have negative implications on the infection susceptibility of older adults [28,29]. In light of these findings, it is not surprising that several age-related disorders are directly related to pulmonary pathologies. In particular, it is estimated that frailty syndrome affects one in five patients with chronic obstructive pulmonary disease (COPD) [30].

Concurrently, neurological disorders are related to a higher risk of pulmonary complications [31,32]. Dysphagia, compromised glottic closure, decrease cough reflex, decreased level of consciousness, mechanical ventilation, respiratory muscle weakness, abnormal respiratory patterns, and an inability to manage airway mucus clearance are common pathological conditions promoting pulmonary complications [33,34,35,36].

Similarly, frail patients with cardiologic disorders might frequently be affected by pulmonary diseases due to the strict linking between respiratory and cardiological systems, systemic inflammation, and the effects on vascular endothelium and coagulation pathways [37]. Moreover, several common risk factors and negative lifestyle behaviors might affect both pulmonary and cardiocirculatory systems [37]. Accordingly, cancer patients might frequently experience pulmonary symptoms affecting both performance status and health-related quality of life (HR-QoL) [38,39,40], and to date, dyspnea is among the most common symptoms reported by end-stage cancer patients [41]. Moreover, cancer and its treatment might significantly affect respiratory function, requiring specific support and rehabilitation interventions to manage these burdensome conditions [42,43].

Although PR is a cornerstone of the therapeutic management of COPD [2], poor attention is historically paid to PR in preventing or managing pulmonary complications of frail patients. On the other hand, PR might play a role in addressing the critical issue of the chronicization of disabling conditions and preventing exacerbation of pulmonary disorders with positive implications even in health care costs due to rehospitalization [44]. In light of these considerations, we summarized in Figure 1 the most common disabling conditions that might benefit from PR.

Altogether, these findings highlighted the need for a patient′s tailored approach including PR to prevent and/or reduce disability in patients suffering from any chronic and symptomatic respiratory condition [13]. Moreover, due to its high plasticity and its multidimensional characteristics, PR might target several disabling conditions in a precise approach to frail patients.

In light of these considerations, effective tailored strategies including PR in both inpatient and outpatient settings, or community dwelling settings, should be integrated into a comprehensive rehabilitation approach for frail patients with pulmonary function impairment.

## 4. Integrating Pulmonary Rehabilitation in Inpatient and Outpatient Settings

PR is proposed to improve outcomes in patients with pathological respiratory conditions in several settings, with promising research results, but with several translational concerns in the clinical setting (Figure 2 represents the current therapeutic pathways for patients needing pulmonary rehabilitation).

Indeed, PR still remains unacceptably underused worldwide, with growing evidence underlining that several patients have limited access to PR, while most of them do not complete the rehabilitative programs. An interesting study from Spitzer et al. [46] reported that only 2.7% of US Medicare patients were referred to a PR program within 12 months after a COPD exacerbation. The reasons underpinning these critical data include the lack of healthcare resources (or their inadequate allocation), lack of awareness of patients and/or clinicians, and scarcity of specialized centers of healthcare professionals’) with an adequate specialization and/or training opportunities.

In this context, the American Thoracic Society and the European Respiratory Society published in 2015 a policy statement suggesting strategies to overcome these barriers and support PR implementation in common clinical practice. However, most of these policies rely largely on the choices of local governments and their health resources allocation strategies [47]. While hospital-based models (in-patient and out-patient) were, for many years, the most common way to deliver PR programs, a home-based model seems to be a feasible option for increasing PR delivery among patients who live far from referral centers and/or for those experiencing physical or social limitations [48].

New projects were proposed to increase delivery and uptake of PR in different settings. An ongoing study from Marques et al. [49], planned a 12-week, community-based PR program to engage primary healthcare areas where they are not available, by training HCPs in the main components of PR.

Moreover, another critical issue is positive PR outcome maintenance over time. A recent systematic review showed that supervised maintenance programs after a structured PR protocol for COPD patients could improve health-related quality of life and exercise capacity at six to twelve months, without adverse events or adjunctive effects on exacerbations, hospitalization rate, and mortality [50].

Taken together, these data underline the need for sustainable strategies to bridge the hospital-based only PR delivery to a community/home-based model closer to the patient′s everyday life setting. This could overcome barriers, such as commuting times, economic and psychosocial issues, with potential improvement in access and adherence to these interventions. However, to the best of our knowledge, no previous study focused on organizational models implementing PR delivery in frail patients.

## 5. Sustainable Strategies in the Comprehensive Management of Frail Patients with PR Issues

Although high-quality evidence supports the efficacy of PR in the management of patients suffering from chronic and symptomatic respiratory conditions, several barriers still affect participation in PR programs. Therefore, recent research is now focusing on effective strategies to improve participation in tailored PR programs in order to enhance the positive results already shown in the current literature. In this scenario, recent reports emphasized the need for early identification of frailty, since frail patients might experience faster and greater benefits from PR, but also a steeper decline, and the greatest need for maintenance [51]. Moreover, it should be noted that the presence of comorbidities in patients undergoing PR programs ranged from 50 to 60% when self-reported, and up to 97% when objectively assessed [52]. Therefore, it is mandatory to develop inclusive and easily accessible organizational models for frail patients needing PR interventions.

Although the evidence aiming at improving PR program participation mainly focused on COPD patients, a translational approach should be considered in order to fill the gap in organizational models promoting PR programs in frail patients with other disabling diseases. More in detail, the recent systematic review by Robinson et al. [53] underlined that continued support from HCPs or continued peer interaction, continuous feedback, self-monitoring, and participation in physical activity groups might represent key facilitators to optimize patient participation and adherence to PR programs. In contrast, anxiety and fear, breathlessness upon exertion, restricted access to social support and structured maintenance sessions, and lack of positive feedback from HCPs might be considered the most common barriers to improve participation in PR programs [53]. Moreover, environmental factors might drastically affect PR service accessibility. In particular, despite PR inpatients′ intensive services being effectively distributed [47], the outpatient referral might be lacking in efficiency, with detrimental consequences in referral and uptake rates that are low worldwide [47,52]. In this context, limited knowledge by the health practitioners of PR programs might be partly related to the gap between outpatient and inpatient settings, with negative implications for the optimal directing of patients toward the outpatient services. Moreover, patients’ access difficulties (both practical and self-perceived) might negatively affect the PR program′s effectiveness in the outpatient settings [52].

As a result, networking between hospital staff and HCPs in community settings represents a critical issue in the current literature [54]. Regrettably, specific organizational models addressing sustainable PR strategies to treat frail patients are lacking. However, telemonitoring systems are currently being proposed to precisely address the homecare needs of frail patients with potential advances in hospital and home care networking promoting both healthcare and PR delivery at home [55].

Altogether, these findings underlined that PR programs should be tailored to the specific patient condition, taking into account the personal physical and psychosocial status and the environmental characteristics [56]. Interestingly, in recent years, growing attention is rising on innovative approaches to face the problem of the availability of PR programs, including the implementation of new technologies, telemedicine, telemonitoring, and virtual reality tools to improve the management of frail patients and focus resources on the long-term management of these chronic conditions [57].

## 6. Telemedicine and Technological Innovation to Overcome the Barriers to PR

During the COVID-19 pandemic, several frail patients became isolated at home with detrimental consequences in terms of their functional status and social participation [58,59]. Therefore, in recent years a growing interest in telehealth and telerehabilitation solutions that might improve healthcare delivery in frail patients is observed [60,61,62].

To date, several telemedicine programs were recently proposed to improve both screening and therapeutic intervention in frail patients suffering from pathological respiratory conditions [60,61,62]. In particular, telehealth exercise interventions and digitally supported territorial health networks were effectively introduced to overcome barriers to in-person medical assistance [60,61,62].

Interestingly, in a recent RCT by Rutkowski et al. [63], virtual was proposed as a promising tool for boosting PR benefits in patients with chronic pulmonary diseases. Although the quality of evidence is still low, all those techniques that include exercise training, education, and behavior change are shown to be as safe and effective for the improving exercise capacity and health-related quality of life as other forms of PR delivery [24,64,65,66].

Despite several authors suggesting telerehabilitation be used as a potential add-on to overcome the barriers to PR delivery, the optimal training duration is far from being fully characterized, and several studies assessed PR programs from five times [63,64] to seven times a week [64,67]. In addition, there is no evidence about the optimal strategy for integrating telerehabilitation with standard PR protocols in frail patients.

On the other hand, the safety and validity of web-based or telerehabilitation models are still debated. A recent review reported no significant differences between standard PR and telehealth PR in terms of quality of life and exercise capacity improvement [68]. Thus, clinical implementation of telehealth PR might be considered a suitable option to optimize health care delivery and minimize sanitary costs. To date, it is widely accepted that standard PR is a cost-effective therapy, reducing health care resources in the long-term management of COPD patients [69]. Thus, telerehabilitation might further reduce costs due to the in-person assessment and overcoming distances. Therefore, telehealth and telerehabilitation solutions might be effectively integrated into the sustainable management of frail patients needing PR, especially those with functional impairment and impaired mobilization that affects accessibility to standard PR programs. Despite these considerations, several limitations might reduce engagement in telerehabilitation programs in older subjects, including lack of access to technology or scarce technology skills, or a patients’ impairment of hearing, vision, communication, or cognitive functions [70,71,72]. Therefore, caregivers might play a crucial role and might be instructed by HCPs to provide the optimal assistance in telerehabilitation delivery [73].

In this context, the strategies to increase the awareness of both caregivers and HCPs might begin with specific training courses, that should teach the rationale and indications of PR. Therefore, universities might have a key role in the implementation of health care and telehealth care skills developing training courses and specific didactical paths in order to improve HCPs knowledge of PR for frail patients in clinical settings.

## 7. The Role of Training Healthcare Professionals and Patients’ Engagement

In recent years, growing evidence emphasized the crucial role of patients’ engagement in improving satisfaction on medical treatment and cost-effectiveness in service delivery [74]. Despite the emerging research that is now focused on tailored approaches or innovative technologies to enhance a patients’ engagement [75,76], a key role needs to be played by general practitioners and all HCPs involved in the management of patients with chronic respiratory diseases. In this scenario, a recent review including 26 studies assessed the barriers affecting patients’ engagement, reporting that patients’ participation might be severely affected by lack of training and HCPs uncertainty about how to deal with patients and act on their feedback [77]. Therefore, effective strategies improving patients’ engagement and a patient-centered care practice should include HCPs training courses about the components and benefits of PR, in order to adequately engage the patient in an appropriate evidence-based care process.

Moreover, in recent years, there was a growing population of patients with multiple chronic conditions and/or requiring long-term oxygen supplementation, devices for respiratory support (CPAP or mechanical ventilation), airway clearance (portable suction unit, cough machines, and other mechanical devices), nutrition, and medication [78]. In this context, emerging technologies and telemonitoring systems were recently proposed to enhance a patient′s centered approach based on community care empowerment and the early detention of exacerbations [79,80,81]. However, several concerns are still open about both community clinicians’ and HCPs’ awareness about technology solutions in order to meet the needs of patients.

These concerns might be partly related to educational training for HCPs that is still largely heterogeneous from short professional bachelor’s degree to academic Master of Science. Curricula can differ among countries, even among universities, and specific training on essential components of PR is not warranted.

Physiotherapists in many countries can count on a growing number of postgraduate training courses and vocational master’s degrees in respiratory physiotherapy. In 2014, the respiratory and physiotherapy task force of the European Respiratory Society published the harmonized core syllabus for postgraduate training in respiratory physiotherapy, covering “the wide range of patients, pathologies, and settings a respiratory physiotherapist can be involved in” [82]. However, a specific postgraduate training for other components of the multidisciplinary teams is still lacking. The recent review indicated a lack of standardized telehealth education competencies [83]. These foundational competencies might include terminology, definitions, technologies applications, health informatics integration, legislation and policy, credentialing and privileging, regulations related to the professional scope of practice, and especially practical skills [83]. In order to increase nursing competencies in telemedicine, the telehealth education should be included in the nursing curriculums [84].

Besides these considerations, a growing amount of literature focuses on the crucial role of the interdisciplinary management of frail patients [85,86,87]. Therefore, a transversal knowledge of the basics of PR in all HCPs and basic medicine might improve the complex management of these patients. In light of these considerations, we summarized in Figure 3 an organizational model proposal based on the before mentioned evidence, highlighting the role of interdisciplinary care in several settings for patients needing PR. Moreover, we integrated the most recent strategies proposed to overcome barriers to the PR management of frail patients.

Besides its limitations, this review sought to be a catalyst for future research implementing the engagement of patients suffering respiratory comorbidities and improving awareness of HCPs in an effective and safe therapy, which is unfortunately still underestimated, but that can significantly impact the disease path of frail elderly patients.

## 8. Conclusions

Taken together, several barriers currently affect the effective integration of PR in the therapeutic management of frail patients in both inpatient and outpatient settings. Sustainable strategies are mandatory to improve the respiratory function and quality of life of these patients, not only to reduce complications and hospitalization rates, but also to decrease sanitary costs related to secondary pulmonary diseases. In this context, promising features were proposed for digital innovation and telemedicine for an integrated program aiming at systematically monitoring and enhancing the early management of frail patients.

Moreover, an interdisciplinary approach including different HCPs might play a role in the comprehensive management of complex patients who experience several pathologies with multilevel interactions.

Unfortunately, additional efforts are required to increase both patients’ and HCPs’ awareness of the need to implement pulmonary rehabilitation in the conventional management of frail patients.

## Figures and Tables

**Figure 1 ijerph-19-09150-f001:**
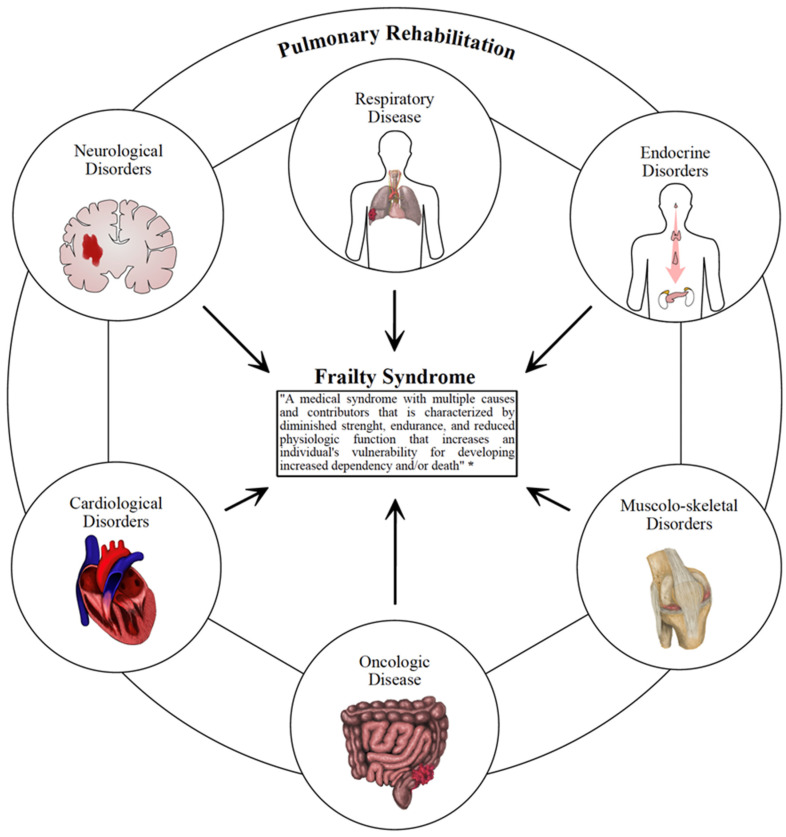
This figure summarizes the most common disabling conditions that might benefit from pulmonary rehabilitation. * Morley JE et al. [45].

**Figure 2 ijerph-19-09150-f002:**
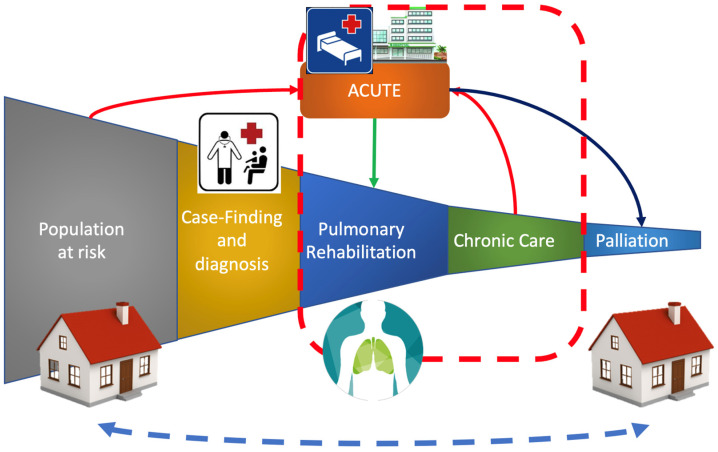
Classic therapeutic pathways for patients needing pulmonary rehabilitation.

**Figure 3 ijerph-19-09150-f003:**
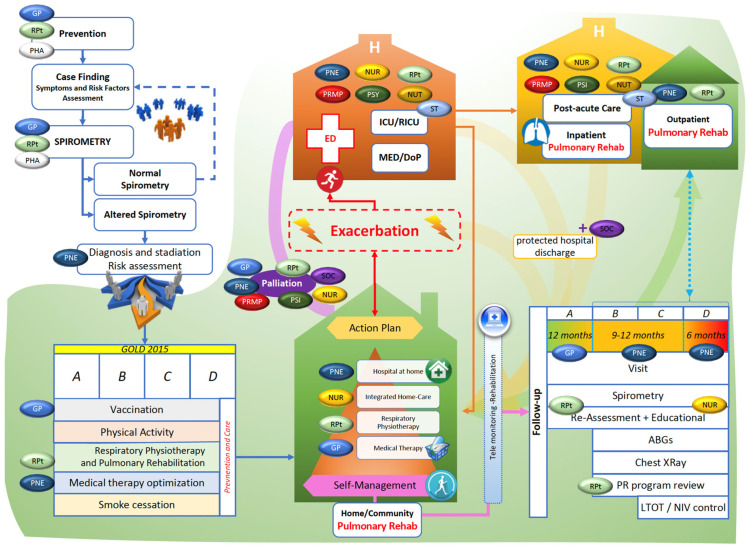
Organizational model proposal, based on chronic respiratory disease evidence, for improving the clinical management of frail patients. Abbreviations: DoP: department of pulmonology; ED: emergency department; GP: general practitioner; ICU: intensive care unit; MED: medical department; NUR: nurse; NUT: nutritionist; PHA: pharmacologist; PSY: psychologist; PNE: pneumologist; PRMP: physical and rehabilitative medicine physicians; RICU: respiratory intermediate care unit; RPt: respiratory physiotherapist; ST: speech therapist; SOC: social assistant.

## Data Availability

The datasets generated during the current study are available from the corresponding author on reasonable request.

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
