# Peer review of "Closing the Gap between Inpatient and Outpatient Settings: Integrating Pulmonary Rehabilitation and Technological Advances in the Comprehensive Management of Frail Patients"

_ijerph, 2022, doi:10.3390/ijerph19159150_

Round 1

Reviewer 1 Report

Your work is a very important statement because the time of hospitalisation was running shorter during the last years.

Please discuss possibilities of networking between the staff of the hospital and the people working out of the hospital.

Is there an indication for medication to improve motivation of frail patients at home especially when they are living alone?

I like your concept of telemedicine and usage of technical devices to overcome the barriers to PR. But which kind of education is necessary and do you need staff to explain usage of these technologies.

And last comment: Who comes on weekend at home and in the clinic, training should be done 7 days or not? Is now the time for the robot to care at home?

Author Response

Dear Reviewer

thank you for your letter and kind comments concerning our manuscript entitled “Closing the Gap Between Inpatient and Outpatient Setting: Integrating Pulmonary Rehabilitation and Technological Advances in the Comprehensive Management of Frail Patients”. We would like to express our sincere appreciation for your careful reviewing and invaluable comments which help us to further improve this paper.

Revisions based on your comments are highlighted in the manuscript in yellow, and our detailed responses according to each revision are shown as followed.

Please discuss possibilities of networking between the staff of the hospital and the people working out of the hospital.

We would like to thank the reviewer for his/her insightful comment. We clarified the needing for a specific organizational model promoting efficient networking between hospital staff and HCPs in the community to better address the needs of frail patients at home.

Is there an indication for medication to improve motivation of frail patients at home especially when they are living alone?

We would like to thank the reviewer for his/her insightful comment.

I like your concept of telemedicine and usage of technical devices to overcome the barriers to PR. But which kind of education is necessary and do you need staff to explain usage of these technologies.

We would like to thank the reviewer for his/her insightful comment. We better characterized the components of standardized telehealth education in accordance with the Reviewer's instructions. In particular, the foundational competencies that should be included in HCPs training courses might include terminology, definitions, technologies applications, health informatics integration, legislation and policy, credentialing and privileging, regulations related to professional scope of practice, and especially practical skills.

And last comment: Who comes on weekend at home and in the clinic, training should be done 7 days or not? Is now the time for the robot to care at home?

We would like to thank the reviewer for his/her insightful comment. The reviewer clearly highlighted a large gap of knowledge; we further improved the manuscript better emphasizing this issue. In particular, the optimal training duration is far from being fully characterized with several studies assessing PR programs 5 to 7 times a week. In addition, there is no evidence about the optimal strategies integrating telerehabilitation into standard PR in frail patients. We further improved this paper by better addressing this critical issue in accordance with the Reviewer’s instructions.

Reviewer 2 Report

Dear Authors,

this is a well-written manuscript presenting the benefits of pulmonary rehabilitation in frail patients and the current challenges, regarding the practical implementation of rehabilitation programs in out- and inpatient medical settings. Your review summarizes in a comprehensive and understandable way the positive effect of pulmonary rehabilitation in frailty and the factors that may influence the patients’ access and adherence to rehabilitation programs and subsequently compliance, satisfaction and effect maintenance. The literature sources you used came frequently from the field of COPD. This may be considered as legitimate, as pulmonary rehabilitation research is mainly based on this patients’ group and COPD is considered to be a systemic chronic disease, which is connected to frailty. The novel aspects of telemedicine and support through new-edge technology seems very interesting. The figures you provided are very illustrative and contribute an additional value to the text. Please pay attention to the following points:

Line 77: please explain the abbreviation RCTs

Line 266: please explain the abbreviation HRPs

Line 313: you are referring to Figure 3 and not Figure 2, please correct

Figure 3: please correct the word Diagnosys as Diagnosis in the left column of the figure

Figure 3: please correct the word mesi (Italian) as months in the right column of the figure

With Best Regards

Author Response

Dear Reviewer

thank you for your letter and kind comments concerning our manuscript entitled “Closing the Gap Between Inpatient and Outpatient Setting: Integrating Pulmonary Rehabilitation and Technological Advances in the Comprehensive Management of Frail Patients”. We would like to express our sincere appreciation for your careful reviewing and invaluable comments which help us to further improve this paper.

Revisions based on your comments are highlighted in the manuscript in yellow, and our detailed responses according to each revision are shown as followed.

Line 77: please explain the abbreviation RCTs

We would like to thank the reviewer for his/her insightful comment. The abbreviation of RCTs has been clarified in the text.

Line 266: please explain the abbreviation HRPs

We would like to thank the reviewer for his/her insightful comment. We corrected the typing error (HCPs instead of HRPs).

Line 313: you are referring to Figure 3 and not Figure 2, please correct

We would like to thank the reviewer for his/her insightful comment. We corrected the typing error following the Reviewer's instructions.

Figure 3: please correct the word Diagnosys as Diagnosis in the left column of the figure

We would like to thank the reviewer for his/her insightful comment. We corrected the typing error in the figure.

Figure 3: please correct the word mesi (Italian) as months in the right column of the figure

We would like to thank the reviewer for his/her insightful comment. We improved the Figure following the Reviewer's instructions.
